# Aurora Classification in All-Sky Images via CNN–Transformer

**Jian Lian** [1] , **Tianyu Liu** [2,*] **and Yanan Zhou** [3,*]

1   School of Intelligence Engineering, Shandong Management University, Jinan 250357, China; 14438120200681@sdmu.edu.cn
2   School of Business Administration, Shandong Management University, Jinan 250357, China
3   School of Arts, Beijing Foreign Studies University, Beijing 100089, China
*   Correspondence: 14438120200291@sdmu.edu.cn (T.L.); zhouyanan@bfsu.edu.cn (Y.Z.)

**Abstract:** An aurora is a unique geophysical phenomenon with polar characteristics that can be directly observed with the naked eye. It is the most concentrated manifestation of solar–terrestrial physical processes (especially magnetospheric–ionospheric interactions) in polar regions and is also the best window for studying solar storms. Due to the rich morphological information in aurora images, people are paying more and more attention to studying aurora phenomena from the perspective of images. Recently, some machine learning and deep learning methods have been applied to this field and have achieved preliminary results. However, due to the limitations of these learning models, they still need to meet the requirements for the classification and prediction of auroral images regarding recognition accuracy. In order to solve this problem, this study introduces a convolutional neural network transformer solution based on vision transformers. Comparative experiments show that the proposed method can effectively improve the accuracy of aurora image classification, and its performance has exceeded that of state-of-the-art deep learning methods. The experimental results show that the algorithm presented in this study is an effective instrument for classifying auroral images and can provide practical assistance for related research.

**Keywords:** auroral image classification; machine vision; deep learning





## 1. Introduction

An aurora is an illustrative embodiment of the coupling between the solar wind and the magnetosphere [1]. It is a light-excitation phenomenon that occurs when high-energy charged particles moving along the magnetic field line settle to the height of the polar ionosphere to excite atmospheric particles. Some high-energy charged particles [2] are generated by the magneto-dynamic process of the solar wind–magnetosphere interaction [3], and the solar wind directly carries others [4]. Observing the morphology of auroras can give sufficient information about the magnetosphere and solar–terrestrial electromagnetic activities in space, which is beneficial for an in-depth understanding of the way and extent of solar activities affecting the Earth, and is also of great significance for mastering the changing laws of space weather processes [5]. The variable auroral morphology is the reaction of a particular dynamic process of the magnetosphere and ionosphere to the atmosphere.

Relevant research has shown that different types of auroral morphology are related to certain magnetospheric boundary layer dynamics processes and that changes in solar wind parameters directly affect the morphology of auroras [1]. Through an all-sky observatory, a high-resolution auroral imaging device, vast amounts of morphological information about auroras can be obtained. Accordingly, the study of auroral phenomena from the perspective of images has received increasing attention. However, most of the recent research relies on the human visual system, which is laborious and time-consuming. Therefore, dealing with image samples reaching up to 20TB annually [6] in an automated fashion has become a challenging task for aurora researchers globally.

Many traditional machine learning-based methods have been proposed to assist in the task of auroral image classification. Yet, the performance of the machine learning algorithms depends on manually crafted features and an appropriate classifier. For instance, in the early work in this area presented by Syrjäsuo and Pulkkinen [7], all-sky images collected by the Finnish Meteorological Institute were leveraged, and the authors developed a machine vision algorithm to classify the images using skeletons of the auroral morphology within a noisy environment. To research the near-Earth space, Syrjäsuo and Donovan [8] introduced a K-nearest neighbor (KNN) algorithm to implement aurora detection in an image and achieved an accuracy of around 90%. In 2011, Syrjäsuo and Partamies [6] detected the aurora's existence, a fundamental task for auroral image analysis and processing. This study also represents an early work in auroral image classification with the moon present. To address four-class all-sky auroral image recognition, Yang et al. [9] employed the features of both spatial texture and a hidden Markov model (HMM). In this study, a set of uniform local binary patterns was used to represent the spatial structures of the images. In 2013, an auroral image classification approach was proposed by [10]. By incorporating linear discriminant analysis (LDA) and saliency information, the proposed method could provide semantic information within the auroral images. In addition, a support vector machine (SVM) was applied to perform auroral image classification. Rao et al. [11] proposed an approach for automatically classifying all-sky images into three categories, including aurora, none-aurora, and cloudy. A variety of features were extracted from the images and classified using SVM. The experimental outcome demonstrated that one specific type of scale-invariant feature transform (SIFT) feature achieved superior performance over the other types of features. Note that the above-mentioned machine learning-based classifiers have insufficient descriptive ability due to the employment of hand-crafted features, which might not be suitable for the requirements of auroral classification.

Deep learning-based models have also shown promising outcomes in auroral image classification. For instance, Clausen and Nickisch [1] introduced a pre-trained deep learning network for extracting the features of 1,001 dimensions from the images; a ridge classifier was trained using the extracted features with an accuracy of 82%. In this study, the authors first labeled 5,824 Time History of Events and Macroscale Interactions during Substorms (THEMIS) [12] images into six types: arc, diffuse, discrete, cloudy, moon, and no-aurora. These six types can also be categorized into aurora and no-aurora, which can be used to train the binary classifiers. A weakly supervised pixel-wise image classification method was proposed in the work of [13]. Zhong et al. [14] proposed an auroral image classification model for polar research based on deep learning networks. In this study, three typical deep learning models were leveraged, including VGG [15], ResNet [16], and AlexNet [17], without manual interventions. The experiments demonstrated the effectiveness of this pipeline. Yang and Zhang [18] used convolutional neural networks to implement an end-to-end auroral image classification framework with four types of auroral images, including arc, drapery corona, radial corona, and hotspot corona. Sado et al. [19] developed an algorithm for transfer learning for auroral image classification. Accordingly, the authors evaluated the performance of 80 neural networks using the six-class Oslo Auroral THEMIS (OATH) dataset [1]. Using the deep learning models as the feature extractors, an SVM classifier was attached to the last layer of the optimal feature extractor. Recently, Yang, Wang, and Ren [20] presented a few-shot learning algorithm for auroral image classification with samples without ample labels. In addition, a cosine classifier was exploited in this study to decrease over-fitting issues. Since most of the above-mentioned methods still rely on the features extracted by convolutional layers, especially in convolutional neural network (CNN)-based frameworks, some studies have explored the employment of transformer models in auroral image classification. For instance, Shang et al. [21] evaluated the performance of both CNNs and transformer models for classifying the auroral images. Note that the performance of CNNs is constrained by the employment of local receptive fields, which could be addressed by introducing transformers.

Bearing the above analysis in mind, we proposed a CNN–transformer model by introducing vision transformers, which are a type of novel deep learning model based on the CNN model [22] and the vision transformer [23] for auroral image classification. In the presented model, both the information in the local receptive field and the global receptive field of the images could be employed. In addition, the attention mechanism was supposed to unveil the global associations between the long-distance pixels in the auroral images. Furthermore, to evaluate the performance of the presented method, we conducted comparison experiments between the state-of-the-art algorithms and ours. The experimental results demonstrated that the proposed algorithm achieved superior performance over the competing techniques in terms of sensitivity, specificity, and accuracy. To compare with the other works in a fair fashion, we leveraged the original OATH dataset [1] with six classes of auroral images, including arc, diffuse, discrete, moon, cloudy, and no-aurora. Both the binary classification (aurora and no-aurora) and hexagonal classification (arc, diffuse, discrete, moon, cloudy, and no-aurora) were implemented by using the presented methods in the comparison experiments. In general, the contribution of this study includes: (1) This is an early work using a CNN–transformer model in automatic auroral image classification; (2) Both the local receptive field and global receptive field of auroral images can be utilized by the proposed model; (3) The proposed model achieved superior performance over the state-of-the-art algorithms.

The remainder of this manuscript is organized as follows: First of all, the dataset used and the details of the presented deep learning models are provided in Section 2; Section 3 describes the experimental outcomes of the state-of-the-art vision transformers on the OATH dataset [1]; the discussion and conclusion of the methods and the outcome from the experiments are provided in Section 4.

## 2. Materials and Methods

This study aimed at determining the optimal transformer model for auroral image classification in an automated fashion. Note that the global associations between long-range pixels in an auroral image can be extracted using the attention mechanism [24]. In addition, to guarantee that the state-of-the-art algorithms could perform a variety of comparisons in a fair manner, we leveraged a publicly available dataset with the default setting of image categories.

All of the related algorithms follow a training–testing strategy. To be specific, the entire dataset is first divided into a training set and a testing set. Furthermore, the presented methods can also be applied to other datasets in a training–testing manner.

### 2.1. Dataset

In the following experiments, the OATH dataset [1] was leveraged, which was constructed by using the THEMIS [12] all-sky images [25]. In the work of [21], there are more categories (eight classes) than the original setting of the OATH dataset, including arc, block, border, cloudy, diffuse, discrete, faint, moon, and others. However, we adopted the OATH dataset's original setting, which includes six classes of labels that could cover the phenomena within ground auroral images (as shown in Figure 1. This choice was made partly to make comparisons between auroral image classification methods fair and partly to constrain the ambiguity of the labels for auroral images.

In total, there were 5824 images randomly selected from the THEMIS all-sky network in the OATH dataset. Following the pre-processing, which was performed as in the work of [1], the auroral images were cropped by 15% to remove irrelevant pixels within an image. Meanwhile, the brightness of the images was scaled to a value between 0 and 1. Details about the image samples are provided in Table 1.

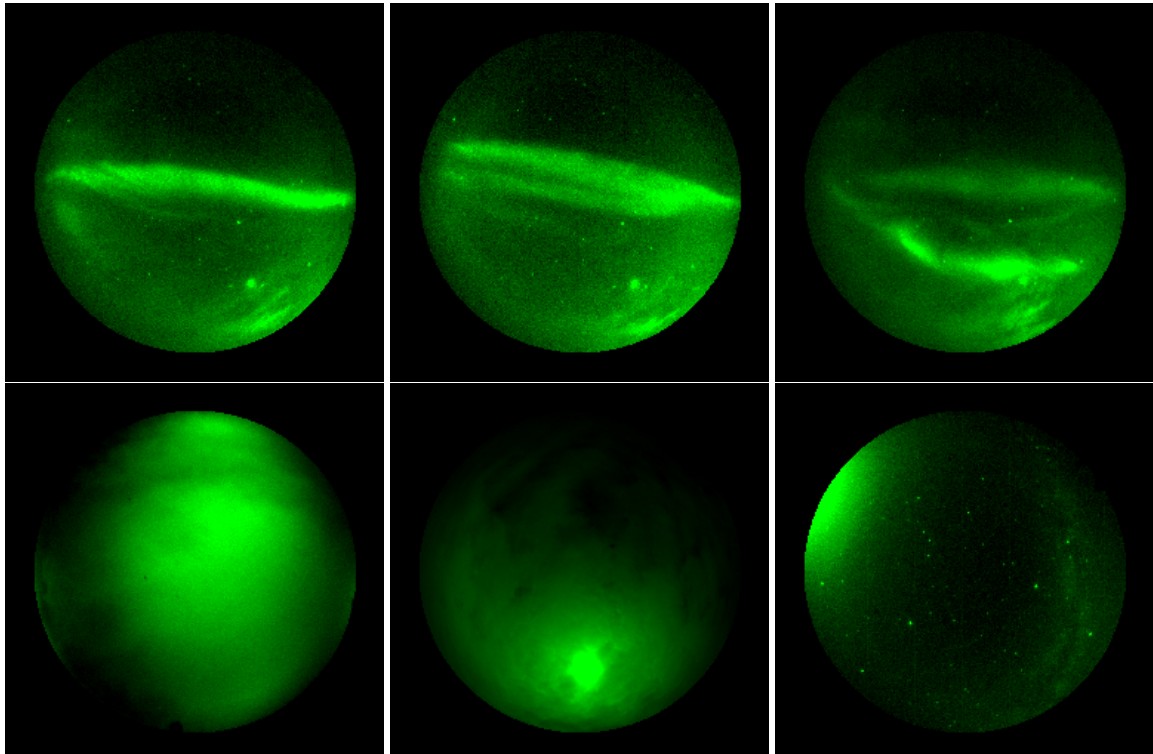

**Figure 1.** Samples of the images in OATH dataset [1]. (**Top left**) arc; (**top middle**) diffuse; (**top right**) discrete; (**bottom left**) cloudy; (**bottom middle**) moon; (**bottom right**) no-aurora.

**Table 1.** Details about the OATH dataset [1].

| Class | Quantity | Hexagonal Classification | Binary Classification |
|---|---|---|---|
| arc | 774 | 0 | 0 |
| diffuse | 1102 | 1 | 0 |
| discrete | 1400 | 2 | 0 |
| cloudy | 852 | 3 | 1 |
| moon | 614 | 4 | 1 |
| no-aurora | 1082 | 5 | 1 |
| Total | 5824 | - | - |

*2.2. CNN–Transformer Model*

In this study, a combination of a CNN and a vision transformer was proposed to address auroral image classification. The presented framework consists of two continuous phases. First of all, one typical CNN architecture, Inception-Resnet-V2 [22], was leveraged as the feature extractor without the back three layers of the original Inception-Resnet-V2 model. Secondly, the outcome from the leveraged Inception-Resnet-V2 in the first phase was taken as the input of the following vision transformer [23]. By dividing the entire OATH dataset into a training set (80%) and a testing set (20%), the proposed model was trained in an end-to-end way. The presented framework is illustrated in Figure 2. To implement the feature extraction for the auroral images, the CNN model Inception-Resnet-V2 first leveraged the stem module, as shown in Figure 3.

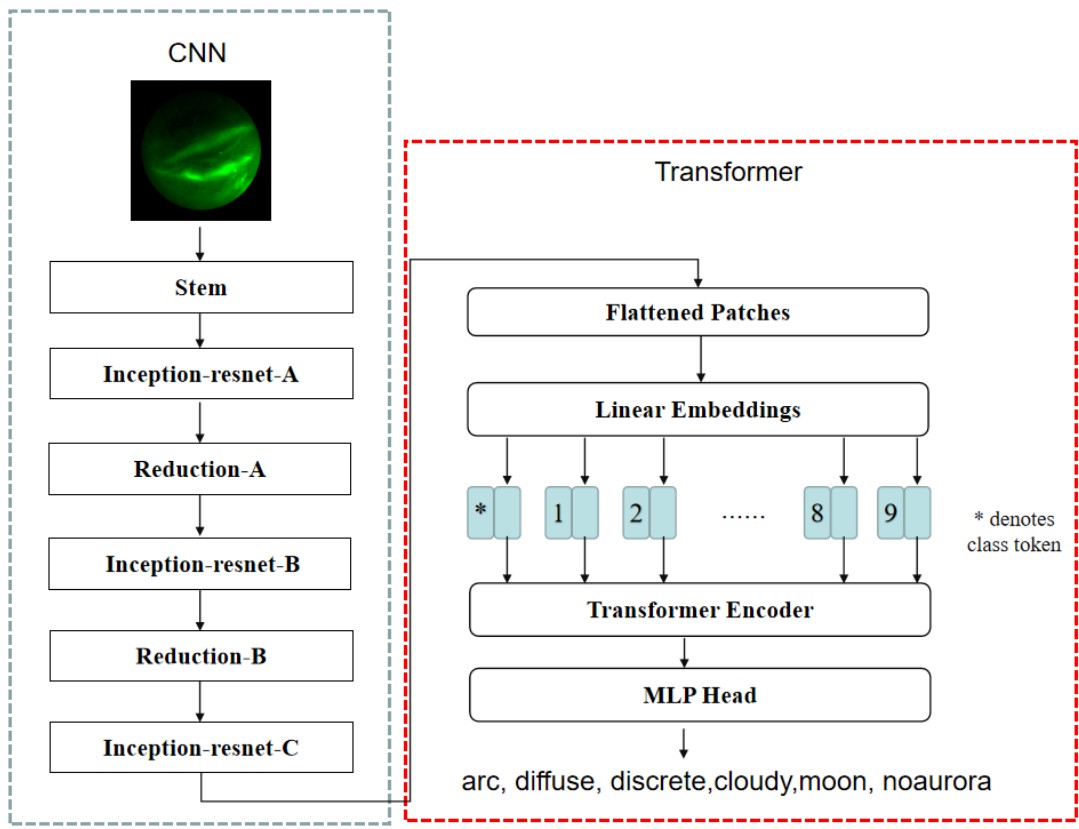

**Figure 2.** The pipeline of the proposed CNN–transformer model.

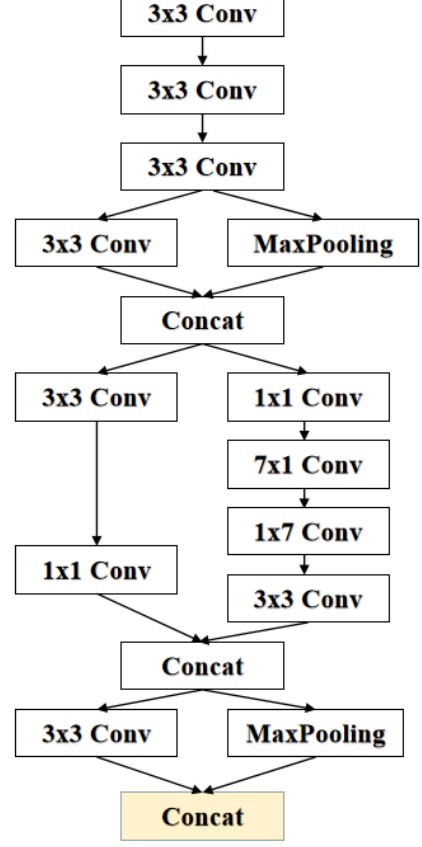

**Figure 3.** The stem module in the proposed Inception-Resnet-V2 model.

As shown in Figure 3, the inner embedding of the input image can be initially extracted using the convolutional operators and max-pooling units in the stem module. Furthermore, the Inception-Resnet-A, Inception-Resnet-B, Inception-Resnet-C, Reduction-A, and Reduction-B modules incorporated in the Inception-Resnet-V2 model were leveraged to refine the features extracted from the stem module. Both of these modules are composed of a group of convolutional operators with various sizes, including $1 \times 1$, $3 \times 3$, $5 \times 5$, $1 \times 7$, and $7 \times 1$. In addition, the $1 \times 1$ operator is supposed to reduce the dimensions of the extracted features. Of note is that with the leveraged inception and residual units in the presented Inception-Resnet-V2 model, both the classification performance and the computation resource efficiency could be guaranteed. Furthermore, the last three layers were removed from the original Inception-Resnet-V2 model since they were exploited classifiers rather than feature extractors.

In addition, the transformer phase in the proposed framework introduced the vision transformer model [23]. The details of the transformer encoder in the presented transformer are provided in Figure 4.

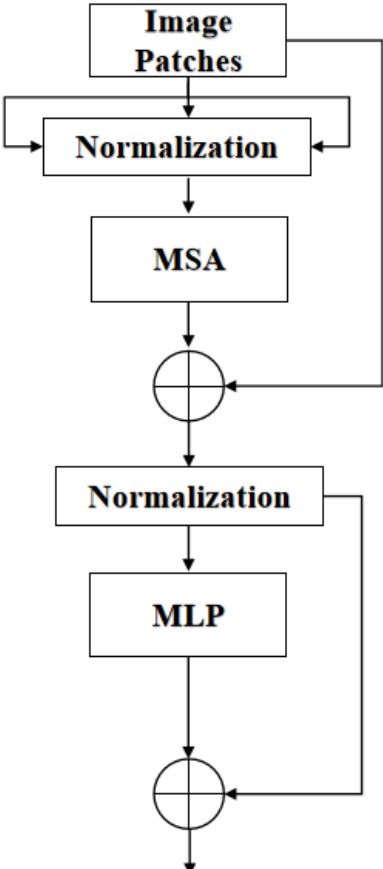

**Figure 4.** The "Transformer Encoder" module in the proposed transformer network.

As shown in Figures 2 and 4, the proposed transformer, inspired by the vision transformer [23] did not incorporate any convolutional modules. To be specific, the input of the proposed transformer was the outcome of the Inception-Resnet-v2 model and was continuously divided into image patches with linear embeddings. The linear embeddings with the class token were then fed into the encoder, as demonstrated in Figure 4, where MLP represents the multi-layer perception module. The encoder module is composed of 16 multi-head self-attention (MSA) modules, MLP modules, and normalization units.

Moreover, the MSA unit is derived from the self-attention mechanism. Both the MLP and MSA operators can be mathematically expressed as Equations (1) and (2):

$$Z'_L = MSA(LayerNorm(Z_{L-1})) + Z_{L-1},\qquad(1)$$

$$Z_L = MLP(LayerNorm(Z'_L)) + Z'_L,\qquad(2)$$

where $L$ represents the number of layers in the MLP module ($L = 16$ in the proposed model), and LayerNorm(.) denotes the operator of normalization.

### 2.3. Training and Fine-Tuning

In addition, the weighing parameters of the proposed CNN–transformer model were first initialized by training on the ImageNet dataset [26]. Then, the proposed framework was trained on the publicly available OATH dataset of six image categories. A random starting strategy was used in the training process, which could decrease the inductive bias and accelerate convergence. The cross-entropy loss function (Equation (3)) was introduced, as shown below.

$$Loss(y, y') = \sum_{i=1}^{C} y_i log(y'_i),\qquad(3)$$

where $y$ represents the ground truth and $y'$ denotes the label prediction.

### 3. Results

### 3.1. Implementation Details and Evaluation Metrics

During the training period, we set the learning rate to 0.001, reduced by 0.5. Three types of image augmentation operations were used, including flipping, rotation, and cropping. The whole process was implemented using Pytorch 1.11 [27] with two NVidia Telsa v100 graphical processing units (GPUs).

To evaluate the performance of the proposed approach and the competing algorithms, the following evaluation metrics were exploited in the experiments: accuracy, sensitivity, and specificity (as shown in Equations (4)–(6)).

$$Accuracy = \frac{TP + TN}{TP + TN + FP + FN},\qquad(4)$$

$$Sensitivity = \frac{TP}{TP + FN},\qquad(5)$$

$$Specificity = \frac{TN}{TN + FP},\qquad(6)$$

where $TP$, $TN$, $FP$, and $FN$ represent true positive, true negative, false positive, and false negative, respectively. The confusion matrix was also used in the evaluation.

### 3.2. Experimental Results

First of all, to evaluate solely the performance of the proposed approach, Figure 5 presents the confusion matrix of the proposed approach on the OATH dataset [1].

In addition, to compare the performance of the proposed approach with the state-of-the-art deep learning models, both the CNNs and transformers were incorporated, including U-Net [28], Mask R-CNN [29], ExtremeNet [30], TensorMask [31], Visual Transformer [32], ViT [23], MViT [33], PVT [34], PiT [35], and Swin Transformer [36]. As demonstrated in Tables 2 and 3, the proposed approach has promising outcomes compared with state-of-the-art techniques. It is notable that both the binary and hexagonal classifications were conducted during the comparison experiment process.

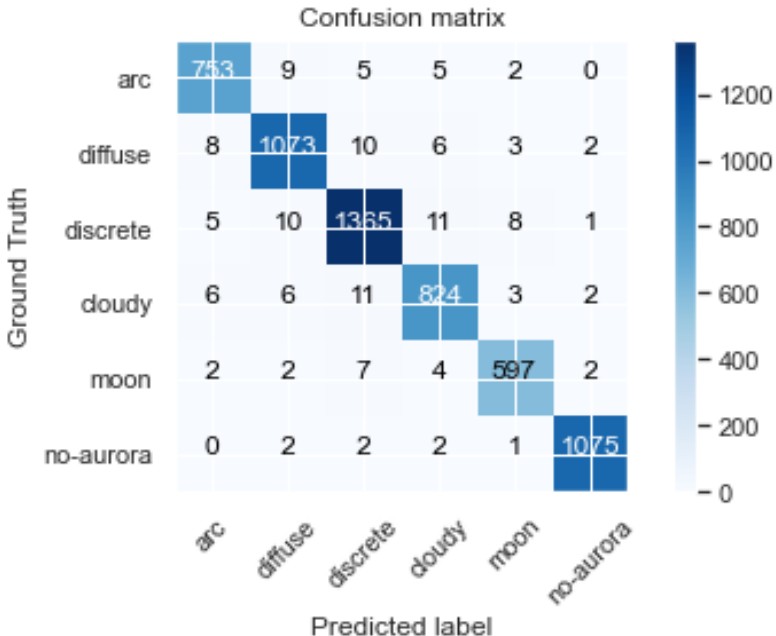

**Figure 5.** The confusion matrix of the proposed method on the OATH dataset [1] of six categories.

**Table 2.** Binary classification (aurora and no-aurora) comparison between the state-of-the-art and the proposed approaches.

| Method | Sensitivity (%) | Specificity (%) | Accuracy (%) |
|---|---|---|---|
| U-Net [28] | 93.2 | 93.8 | 90.8 |
| Mask R-CNN [29] | 92.6 | 92.9 | 89.6 |
| ExtremeNet [30] | 91.7 | 94.7 | 92.1 |
| TensorMask [31] | 92.9 | 93.8 | 94.7 |
| Visual Transformer [32] | 92.6 | 94.2 | 95.0 |
| ViT [23] | 93.7 | 92.7 | 95.2 |
| MViT [33] | 92.4 | 93.5 | 96.3 |
| PVT [34] | 94.7 | 95.3 | 96.1 |
| PiT [35] | 95.4 | 95.5 | 97.8 |
| Swin Transformer [36] | 96.7 | 97.1 | 98.2 |
| The proposed approach | 97.6 | 98.1 | 99.4 |

**Table 3.** Hexagonal classification (arc, diffuse, discrete, cloudy, moon, and no-aurora) comparison between the state-of-the-art and the proposed approaches.

| Method | Sensitivity (%) | Specificity (%) | Accuracy (%) |
|---|---|---|---|
| U-Net [28] | 90.5 | 91.5 | 89.8 |
| Mask R-CNN [29] | 89.7 | 86.1 | 87.8 |
| ExtremeNet [30] | 91.2 | 93.6 | 91.3 |
| TensorMask [31] | 92.3 | 93.1 | 93.7 |
| Visual Transformer [32] | 91.4 | 92.2 | 93.8 |
| ViT [23] | 92.7 | 92.5 | 94.7 |
| MViT [33] | 91.6 | 92.2 | 93.1 |
| PVT [34] | 93.2 | 95.0 | 94.5 |
| PiT [35] | 95.1 | 94.9 | 96.1 |
| Swin Transformer [36] | 96.2 | 96.9 | 97.5 |
| The proposed approach | 97.3 | 98.4 | 98.9 |

Finally, we list the incorrectly classified images in the experiments. To be specific, in Figure 6, the top left image shows that the arc image was classified as a diffuse type; the top right image demonstrates that the discrete aurora was classified as arc; the bottom left image shows that the discrete aurora was classified as diffuse; the bottom right image demonstrates that the cloudy aurora was classified as a moon image.

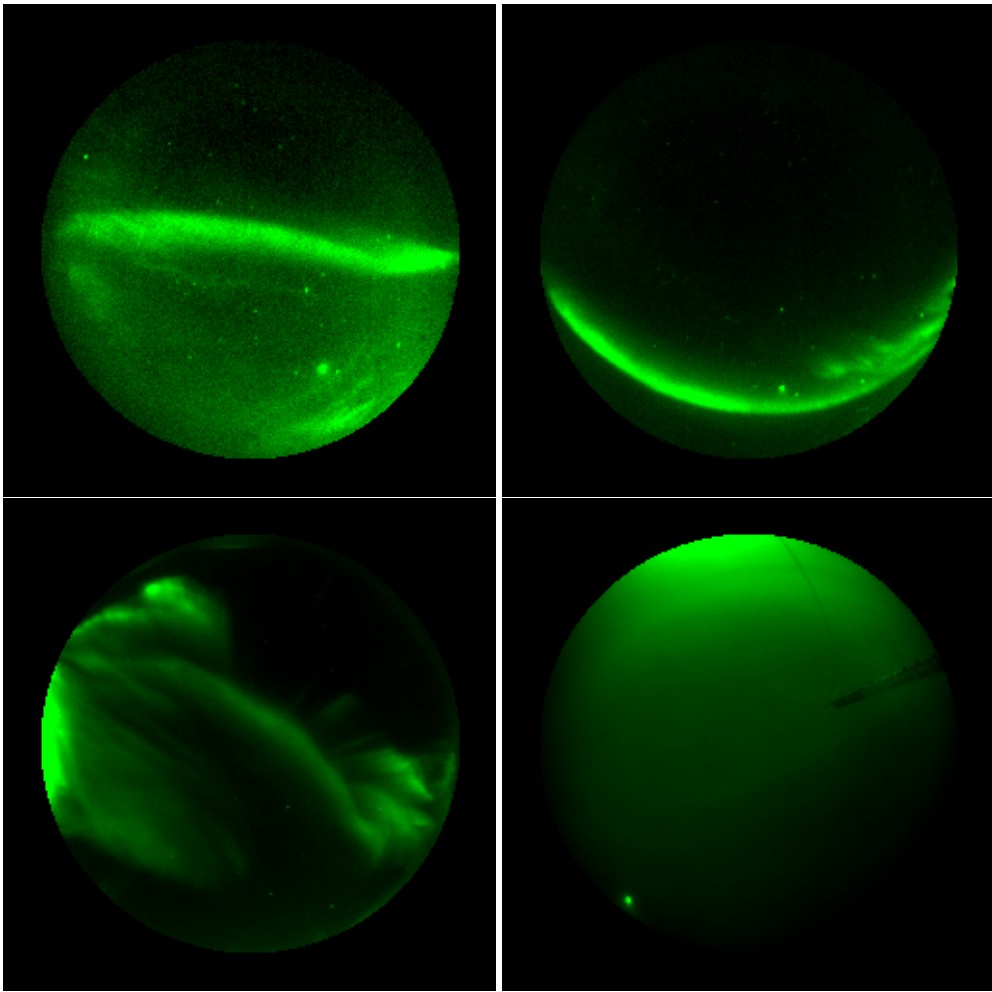

**Figure 6.** The incorrectly classified auroral images in the OATH dataset [1] using the proposed CNN–transformer model.

## 4. Discussion and Conclusions

In this study, we proposed a novel CNN–transformer network for classifying auroral images. In the experiments, the OATH dataset [1]—with 5824 images in six categories, including arc, diffuse, discrete, cloudy, moon, and no-aurora—was used. To evaluate the performance of the proposed approach, both CNN-based and transformer-based models were taken as the competing techniques in the comparison experiments. In general, the proposed approach outperforms both the CNNs and transformers. Therefore, it could be taken as an appropriate instrument for auroral image classification and prediction. Meanwhile, the vision transforms, including Visual Transformer [32], ViT [23], MViT [33], PVT [34], PiT [35], and Swin Transformer [36], achieved superior performances over the CNNs, including U-Net [28], Mask R-CNN [29], ExtremeNet [30], and TensorMask [31]. The primary reason for this is the incorporation of the global receptive field provided by the attention modules in the vision transformers.

Inspired by the work of [21], we further introduced the vision transformer into the classification of auroral images by combining the convolutional modules in the CNN and the MSA units in the transformer models. Accordingly, both the local receptive field

and global receptive field from the auroral images can be fully exploited to enhance the performance of image classification. Moreover, the experimental results demonstrate the effectiveness of the overall CNN–transformer architecture.

This study has the following disadvantages. First, only a publicly available dataset was exploited in the experiments, so a private dataset with more auroral images and more types of auroras should be considered. Second, the transformer model used in the CNN–transformer architecture increased the resource cost of the proposed approach, which makes the proposed approach challenging to apply in practical applications. Finally, as shown in Figure 1, several limitations of the used dataset include the classes of images being poorly defined, the classes being ambiguous, and that there is almost always light pollution from nearby settlements. The quality of the adopted dataset constrains the performance of the proposed model.

In the future, to satisfy the requirements of practical applications, more types of auroras in images and mechanisms such as transfer learning will be incorporated into studies.

**Author Contributions:** Conceptualization, J.L. and T.L.; methodology, J.L.; software, J.L.; validation, Y.Z., J.L. and T.L.; formal analysis, Y.Z.; investigation, J.L.; resources, Y.Z.; data curation, J.L.; writing—original draft preparation, J.L.; writing—review and editing, T.L.; visualization, J.L.; supervision, T.L. and Y.Z.; project administration, T.L.; funding acquisition, J.L. All authors have read and agreed to the published version of the manuscript.

**Funding:** This research was funded by Natural Science Foundation of Shandong Province in China, grant number ZR2020MF133.

**Institutional Review Board Statement:** Not applicable.

**Data Availability Statement:** The data presented in this study are openly available in the OATH dataset at https://doi.org/10.1029/2018JA025274, reference number [1].

**Conflicts of Interest:** The authors declare no conflict of interest.

## Abbreviations

The following abbreviations are used in this manuscript:

| | |
|---|---|
| LDA | Linear discriminant analysis |
| SVM | Support vector machine |
| SIFT | Scale-invariant feature transform |
| THEMIS | Time History of Events and Macroscale Interactions during Substorms |
| OATH | Oslo Auroral THEMIS |
| CNN | Convolutional neural network |
| MLP | Multi-layer perceptron |
| MSA | Multi-head self-attention |
| GPU | Graphical processing unit |
| TP | True positive |
| TN | True negative |
| FP | False positive |
| FN | False negative |

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
