# Peer review of "Aurora Classification in All-Sky Images via CNN–Transformer"

_universe, doi:10.3390/universe9050230_

Round 1

Reviewer 1 Report

The article is devoted to an interesting and topical issue — the classification of data from all-sky cameras that observe auroras. The rapid development of the capabilities of neural networks allows them to be used to process such data.
In principle, the structure of the article is traditional for this area.
The authors used the available dataset. Unfortunately, even the examples given by the authors (Fig.1 and Fig.6) illustrate the main problems in such data:
1) Classes are poorly defined (for example, arc - diffuse radiance - discrete)
2) Ambiguity of classes - different objects are present at the same time.
3) There is almost always light pollution from nearby settlements, which affects the slightest cloudiness or fog.
For such a dataset, a multi
label classification would be more obvious.
But even in such a dataset, neural networks can somehow find differences, which is shown by the table given by the authors.
The description of the network used is clearly insufficient. The fact that the combination of CNN + Transformer is understandable, but no more. The main details in the structure of the transformer are not given, it is not even clear how many "heads" are in the block of attention. But the formulas (4 - 6) given in the cited papers are repeated. What is Fig.4 - it is not clear where this block is in Fig.2. This is important because the authors claim that their configuration is better than in the cited papers. We need to understand why. Maybe just a very complex network that "remembered" the entire dataset?
Moreover, it is difficult to reproduce such a configuration, as the authors write in the conclusions, what is the use then?
English needs editing. Even for a non-native speaker it's hard to read. The use of certain words is questionable. Lots of garbage expressions.
The work needs to be improved, mainly in a detailed description of the model, discussion of problems in the dataset, what is its contribution to science.
Some minor remarks are noted in the text to help in revision.

Author Response

--------------------Response to Reviewer # 1--------------------

The article is devoted to an interesting and topical issue — the classification of data from all-sky cameras that observe auroras. The rapid development of the capabilities of neural networks allows them to be used to process such data.
In principle, the structure of the article is traditional for this area.

The authors used the available dataset. Unfortunately, even the examples given by the authors (Fig.1 and Fig.6) illustrate the main problems in such data:
1) Classes are poorly defined (for example, arc - diffuse radiance - discrete)
2) Ambiguity of classes - different objects are present at the same time.
3) There is almost always light pollution from nearby settlements, which affects the slightest cloudiness or fog.
For such a dataset, a multilabel classification would be more obvious.
But even in such a dataset, neural networks can somehow find differences, which is shown by the table given by the authors of this dataset. 

Point# 1: The description of the network used is clearly insufficient. The fact that the combination of CNN + Transformer is understandable, but no more. The main details in the structure of the transformer are not given, it is not even clear how many "heads" are in the block of attention. But the formulas (4 - 6) given in the cited papers are repeated. What is Fig.4 - it is not clear where this block is in Fig.2. This is important because the authors claim that their configuration is better than in the cited papers. We need to understand why. Maybe just a very complex network that "remembered" the entire dataset? 

Answer: We are sorry for the confusion, and thank you so much for pointing these out. Following your suggestion, we added more details about the transformer used in the proposed approach, including the number of layers "L=16" and the number of  "heads=16" in the revised manuscript. To make clear where the block in Fig.4 is in Fig.2, we have also modified the legend of Fig.4 into "The ‘Transformer Encoder’ module in the proposed transformer network." in the revised manuscript.  

Point# 2: Moreover, it is difficult to reproduce such a configuration, as the authors write in the conclusions, what is the use then?

Answer: We are sorry for the confusion and thank you for pointing this out. In the Conclusion section, we described one of the disadvantages of the proposed approach as "Second, the transformer model used in the CNN-Transformer architecture increased the resource cost of the proposed approach, which makes the proposed approach challenging to apply in practical applications." The main implication of this disadvantage is that Transformer models require a considerable amount of computing resources. However, using mechanisms such as transfer learning, we can still apply them into practical cases.

Following your suggestion, we modified the description as "to satisfy the requirements of the practical applications, more types of auroras in images and mechanisms such as transfer learning will be incorporated into further studies." in the Conclusion section of the revised manuscript.

Point# 3: English needs editing. Even for a non-native speaker it's hard to read. The use of certain words is questionable. Lots of garbage expressions.

Answer: Thank you so much for pointing this out. Following your suggestion, we double-checked the use of certain words the garbage expressions in the revised manuscript.

Point# 4: The work needs to be improved, mainly in a detailed description of the model, discussion of problems in the dataset, what is its contribution to science.

 Answer: Thank you so much for pointing these out. First of all, we added more details about the proposed model in the revised manuscript, including the number of layers "L=16" and number of heads "heads=16" in the revised manuscript. Second, we added the discussion of problems in the dataset in the Conclusion section of the revised manuscript as "Finally, as shown in \myfigure{fig:1}, several limitations of the used dataset include classes of images being poorly defined, classes being ambiguous, and there is almost always light pollution from nearby settlements. The quality of the adopted dataset constrains the performance of the proposed model.". Finally, we added the contribution of the proposed model as "In general, the contribution of this study includes: (1) This is an early work of CNN-Transformer model in automatic auroral image classification; (2) Both the local receptive field and global receptive field of the auroral images can be utilized by the proposed model; (3) The proposed model has achieved superior performance over the state-of-the-art algorithms." in the Introduction section of the revised manuscript.

Point# 5: Some minor remarks are noted in the text to help in revision.  

Answer: Thank you so much for pointing these out. Following the remarks in the text, we have modified the use of words and added the details of the model in the revised manuscript.

Reviewer 2 Report

Authors present an algorithm to classify all-globe aurora images. The accuracy of the algorithm is high (99%). So, the technical part of the paper is correct. I have some requirements on the general imposing of this paper: it refers to the data to the OATH dataset [1] but the reference is a paper - I guess that the dataset is an internet resource.

How the all-aurora images have been obtained - would ask a non-expert reader. What are "some high-energy charged particles"? What are "dynamic processes of magnetosphere to atmosphere" - maybe some references? Can the reader see the relation between the whole-globe image and an image as seen from ground?

English needs corrections: "classification classifiers" (line 70)

Finally - what is the outcome of the paper, I mean - how fig. 1 is different from figure 6. Please extend the figure caption 6.    

Author Response

--------------------Response to Reviewer # 2--------------------

Point# 1: Authors present an algorithm to classify all-globe aurora images. The accuracy of the algorithm is high (99%). So, the technical part of the paper is correct. I have some requirements on the general imposing of this paper: it refers to the data to the OATH dataset [1] but the reference is a paper - I guess that the dataset is an internet resource.

Answer: We are sorry for the confusion and thank you for pointing this out. The Oslo Auroral THEMIS dataset (OATH) dataset can be downloaded from the webpage "http://tid.uio.no/plasma/oath/", in which the authors of OATH dataset made a request

"If you use the Oslo Auroral THEMIS dataset, please refer to:

Clausen, L. B. N., & Nickisch, H. (2018). Automatic classification of auroral images from the Oslo Auroral THEMIS (OATH) data set using machine learning. Journal of Geophysical Research: Space Physics, 123, https://doi.org/10.1029/2018JA025274".

Point# 2: How the all-aurora images have been obtained - would ask a non-expert reader. What are "some high-energy charged particles"? What are "dynamic processes of magnetosphere to atmosphere" - maybe some references? Can the reader see the relation between the whole-globe image and an image as seen from ground?

Answer: We are sorry for the confusion and thank you so much for pointing these out. Following your suggestion, we added the following references for "some high-energy charged particles", "dynamic processes of magnetosphere to atmosphere", and the "relation between the whole-globe image and an image as seen from ground" in the revised manuscript.  

  • Seki, S.; Sakurai, T.; Omichi, M.; Saeki, A.; Sakamaki, D. High-Energy Charged Particles 2015. https://doi.org/10.1007/978-4-431-55684-8.
  • Borovsky, J.E.; Valdivia, J.A. The Earth’s magnetosphere: a systems science overview and assessment. Surveys in geophysics 2018, 39, 817-
  • ZHANG, H.; HU, Z.; HU, Y.; ZHOU, C. Calibration and verification of all-sky auroral image
    parameters by star maps. Chinese Journal of Geophysics 2020, 63, 401–411.

Point# 3: English needs corrections: "classification classifiers" (line 70).

 Answer: Thank you so much for pointing this out. Following your suggestion, we double-checked the grammars and typos in the revised manuscript.

Point# 4: Finally - what is the outcome of the paper, I mean - how fig. 1 is different from figure 6. Please extend the figure caption 6.   

Answer: We are sorry for the confusion and thank you so much for pointing these out. The outcome of the paper is provided in both Fig.5, Tab. 2, and Tab.3. Fig.5 demonstrates the confusion matrix of the proposed model on the OATH dataset of 6 classes. Tab.2 provides binary classification results in terms of sensitivity, specificity, and accuracy of the competing methods on the OATH dataset. Tab.3 provides hexagonal classification results in terms of sensitivity, specificity, and accuracy of the competing methods on the OATH dataset. In addition, Fig.1 is used to demonstrate the auroral images in OATH dataset. Fig.6 is used to demonstrate the incorrectly classified auroral images in OATH dataset by using the proposed CNN-Transformer model.

Following your suggestion, we extended the figure caption 6 in the revised manuscript as "The incorrectly classified auroral images in the OATH dataset [1] using the proposed CNN-Transformer model.".

Round 2

Reviewer 1 Report

The authors have improved the text presented. The text is not self-sufficient, for its understanding it is necessary to familiarize yourself with the cited literature. The used structure of the neural network is new, so the article can be published. Practical use has only theoretical meaning, since the model was trained on a problematic dataset.